# An Experimental Assessment of the Water Permeability of Concrete with a Superplasticizer and Admixtures

**DOI:** 10.3390/ma13245624

**Published:** 2020-12-10

**Authors:** Zdzisław Skutnik, Mariusz Sobolewski, Eugeniusz Koda

**Affiliations:** Institute of Civil Engineering, Warsaw University of Life Sciences-SGGW, 02-787 Warsaw, Poland; mariusz.sobolewski7@gmail.com (M.S.); eugeniusz_koda@sggw.edu.pl (E.K.)

**Keywords:** permeability coefficient, flow pump technique, depth penetration test, superplasticizer, admixtures

## Abstract

This study presents a flow pump technique usually used for evaluating the permeability of soils, which was, for first time, applied to measure the water permeability of concrete. Additionally, a new easy-to-apply method to determine permeability is proposed, based on a modification of Valenta’s formula. In the calculations, the apparent air content of concrete mixes was taken into account. An additional purpose of the conducted research was to determine the influence of a new generation of polycarboxylate superplasticizer and chemically active admixtures on the permeability, compressive strength, and other properties of concrete. The following four types of concrete were tested: concrete without admixtures, concrete with an admixture to increase the compressive strength, concrete with a superplasticizer, and concrete containing two admixtures simultaneously. The results showed that the proposed method allows to obtain reliable measurements within a very short period of time. The obtained results confirmed that new method may be very useful in engineering practice, particularly in terms of the watertightness of hydrotechnical concretes and the properties of the concretes used in bridge construction, underground parts of office buildings, or sealed tanks.

## 1. Introduction

Concrete is one of the major construction materials widely used for infrastructure and building construction across the world. The durability of concrete structures is primarily influenced by the movement of both gaseous and liquid substances via the system of pores, which can possibly result in the deterioration of the quality of concrete [1,2,3]. The ability of concrete to withstand the intrusion of these aggressive agents is mainly characterized by permeability. Therefore, the water permeability of concrete can be used to assess its durability. Measuring the water permeability of concrete should be a simple, quick, and accurate process. Quantitative assessment of the water permeability of concrete can be carried out by using various permeability measurement methods on cementitious materials, each of which is suitable for limited specific applications. In general, permeability testing can be divided into two types: direct and indirect. Most direct methods of determining the permeability of porous materials are based on Darcy’s law [4]. The permeability measurement of concrete by the constant gradient method requires complete saturation of the specimen and a long testing period [4,5,6,7]. The establishment of the equilibrium flow conditions is associated with flow over the whole sample volume. Indirect permeability testing methods are faster than direct methods because they are based on measurements obtained under transient flow conditions [4]. Measurements of water permeability are difficult to perform, and the standard methods of water permeability measurement make it very difficult to compare the results obtained from different devices. The test results from different methods often vary a lot, and, as a result, none of the existing methods are widely or fully accepted [4,7,8,9]. This issue is compounded in the early stages by rapid rates of hydration and by slight differences in the kinetics of the reactions between samples [6]. Even when using one method, the variability of the results is high. In addition, the variability of data increases as the permeability decreases [10]. Very little work has been done on the behavior of concrete after it has been permeated by water [11]. Moreover, it is noted that there are no recommended standard values for applied pressure in permeability tests that are accepted by the whole scientific community [7].

The water permeability of permeable and highly permeable concretes can be assessed easily by direct methods. However, the measurement of water permeability has become more complicated as the quality of concrete has improved. Laboratory permeability tests for modern dense concrete or high-performance concrete may meet some difficulties in keeping the steady state flow through the concrete sample, and there is a high risk of leaking around the sample if the surface is not sealed. For concrete with a high density, it is not possible to obtain a measurable quantity of water flow, even over a long period of time [7].

One method that can be adapted to test the permeability of concrete is the flow pump technique used in the study of low-permeability materials [12,13,14,15,16,17,18,19]. The idea of this method invented and developed by Olsen [20,21] is to ensure a constant flow rate through the sample and control of the hydraulic head difference, which stabilizes when the flow becomes steady. Thus, a constant-flow permeability test is far more rapid than a conventional constant head test, and a substantial period of time—more than 40 h in soils with extremely low permeability—is needed to establish the steady state condition required for calculating permeability from Darcy’s law. The advantages of this test arise from the fact that it is much easier to control small flow rates precisely than to measure them accurately. Therefore, the test allows rapid and accurate determination of hydraulic conductivity even at very low flow rates (Figure 1). As can be seen in Figure 1, the flow of water through the sample in the flow pump technique takes place over a precisely defined area. Preparation of the sample for testing is very simple, and the diameter of the sample can vary from 50 to 100 mm depending on the size of the chamber used. In the laboratory, the sample is prepared by pouring the concrete mortar into a mold of an appropriate diameter and height. In order to determine the permeability of concrete in a structure, it is possible to take core samples of a height of approximately 100–150 mm and a diameter of 100 mm. The pressure/volume controllers used for flow pump testing guarantee very accurate measurements of the pressure/volume. The resolution of the volume measurement is 1 mm^3^, while the resolution of the pressure measurement is 1 kPa on display and 0.1 kPa via software.

However, when testing a large volume of weakly permeable concrete, the limitation associated with obtaining a flow rate through the whole sample in this method is a serious problem. The flow pump technique in adjustment to concrete testing consists of maintaining a constant level of hydraulic head and controlling the flow rate under a given pressure. However, when testing a large volume of weakly permeable concrete, the limitation of this method is obtaining a flow rate through the whole sample. This approach makes it possible to reproduce the test conditions that are used in measuring the depth of water penetration under pressure in accordance with EN 12390-8 [22]. Therefore, the values of the permeability coefficient obtained from the different methods, direct and indirect, can be compared. The short testing time by the flow pump technique eliminates the influence of additional factors on the values of the permeability coefficient. It is therefore possible to determine the permeability coefficient of concrete within a specific time period. This can be particularly important when testing the influence of certain additives and when determining the time after which they are most effective. This paper presents experimental results concerning the permeability of concrete in which the influence of a new generation of polycarboxylate superplasticizers and a chemically active admixture to increase the compressive strength was also evaluated. Additives and admixtures are nowadays incorporated into modern concrete to improve its construction performance, but they also influence the pumping behavior of such concrete, which differs significantly from that of conventional concrete, especially in the initial stage [23,24]. The final goal of using chemical admixtures is to achieve a more durable concrete with self-healing ability [25,26]. Superplasticizers participate in retarding cement hydration and fluidity enhancement [27], and it has a good dispersing effect on cement paste, as well as a water reduction ratio up to 38.6% [28].

## 2. Research Materials and Testing Methods

### 2.1. Materials

The four different types of concretes used in this work were made with Portland cement (CEM I 32.5 R) complying with the European Standard EN 197-1 [29], a fossil sand (0–2 mm), and gravel (2–16 mm). One of them represented a conventional concrete mix and three were admixture-modified concrete mixes [30,31]. The aggregate properties and volumes were constant among the mixes, as was the water-to-cement ratio. The mix proportions and the main properties are given in Table 1. As presented in the table, all mixes had a ratio of water to cementitious material of 0.45. The studied concrete was chosen and designed in order to obtain a compressive strength of approximately 40 MPa at 28 days. For this purpose, two admixtures (A and B) were used for the control composition. One was an admixture (A) with dual action: increasing the compressive strength and sealing capability of concrete. Concrete treated with this admixture demonstrated improved corrosion resistance, as this admixture causes binding of calcium hydroxide and fills capillary pores with low-soluble substances. Admixture (A) was based on Portland cement and mineral particles that contribute to the formation of nanocrystals. The deposition of these nanocrystals was conditioned by the presence of water, and the admixture was added into fresh concrete during batching in the mixing plant, thereby becoming an integrated part of the concrete. The superplasticizer (B) used was CHRYSO Fluid Optima 206, based on a modified polycarboxylate. Concrete with two admixtures offers good mechanical and permeable performances for use in typical construction fields.

Seven cubic samples with sides of 150 mm were prepared for each set of tests on concrete: four for the compressive strength tests and three for the water penetration depth tests. To determine the permeable porosity, six cylindrical samples with a diameter of 45 mm and a height of 40 mm were cast from the same batches of concretes. Additionally, four cylindrical samples with a diameter of 100 mm and a height of 100 mm for each set of tests on concrete were prepared for the flow pump tests at a constant hydraulic head.

All specimens were soaked in water at room temperature after preparation for 10 days. Then, the samples were stored in a controlled chamber at a relative humidity (RH) of 95% and a temperature of 20 °C until 28 days of age. This conditioning of the samples was necessary due to the requirements of the admixture (A) and the delay of cement grain hydration by the superplasticizer (B). The goal was to achieve a high degree of hydration of the polyether side chains of the superplasticizer.

In the case of the concrete mixes, their bulk density was tested in accordance with EN 12350-6 [32] and apparent air content by the pressure method according to EN 12350-7 [33]. The hardened concretes were tested at 28 days of age to find the depth of water penetration, the porosity, and the compressive strength (Table 2). The water permeability was also tested at 28 days by a more sensitive, very high-precision device. The porosity of the hardened concretes turned out to be higher by approximately 10% relative to the apparent air content in the concrete mixes. The porosity calculations were done on the basis of the pycnometric and envelope densities. The devices used were an AccuPyc and a GeoPyc from Micromeritics Instrument Corporation, which can measure the pore volume not available for liquids. Therefore, the total porosities were greater than those obtained from methods using liquids in measurements. The methodology of pycnometric research is presented in [34]. Table 2 presents a list of the test results for the concrete mixes and the hardened concretes.

### 2.2. Water Permeability of the Concrete Obtained by the Depth Penetration Method

Each type of concrete was tested for its water permeability in terms of water penetration depth. These tests were performed on three samples for each type of concrete. The testing procedure for measuring the water penetration depth was carried out in accordance with EN 12390-8 [28], where the sample was not pre-saturated. During the test, a water pressure of 500 ± 50 kPa was applied to the bottom molded face of each cubic sample. The appearance of the surfaces of the test sample not exposed to the water pressure was periodically observed. After 72 ± 2 h, the samples were removed from the apparatus and then each sample was split into two halves to measure the water penetration depth. The difference between the two halves was minimal (3 mm on average), and, therefore, the average of the two was taken. The water penetration depth result was taken as the average water penetration depth of the three cubic samples cast from the same batch of concrete and tested at the same time.

It is believed that in non-saturated medium (partially saturated concrete), the water penetration under external hydraulic pressure is associated with sorptivity and unsteady flow [7,8,35,36]. Thus, this method can quantify the transport under pressure gradient and can possibly assess permeability (in the non-stationary regime). The main assumption is that the penetration test is not continued until a steady state of flow is achieved [7]. Some researchers suggest calculating the penetration coefficient of water by applying Darcy’s law and assuming that the flow of water through the concrete pores is stationary and laminar [7,37,38,39,40,41]. However, the presented studies show that the applied pressure level and the time of measurement are sufficient to obtain results that concern a complete saturated state, even in weakly permeable concretes. Additionally, the high density of the tested concretes and the small depth of penetration in comparison to the test area diameter meet the assumption of uniaxial water penetration. Therefore, this approach provides the coefficient of permeability. 

According to the modified Valenta equation cited by Neville [42] and Ramachandran and Beaudoin [7], the penetration coefficient can be calculated using Equation (1) as follows:(1)$$k=\frac{\alpha d^{2}}{2Ht}$$
where *k* is the penetration coefficient identified with the permeability coefficient (m/s), *d* is the water penetration depth at time *t* (m), *α* is the permeable porosity, *t* is the test time (s), and *H* is the hydrostatic head causing the water percolation (m).

Hedegaard and Hansen [37] expressed d as the maximum penetration depth. Ibrahim and Issa [35] proposed to estimate the penetration coefficient based on the average depth instead of the maximum depth of penetration. Then, Equation (1) can be expressed as follows:(2)$$k={\left(\frac{d_{avg}}{d_{\max}}\right)}^{2}\frac{d_{\max}^{2}}{2Ht}=C^{2}\frac{d_{\max}^{2}}{2Ht}$$
where *C* is the ratio of the *d_avg_* to *d*_max_. The average depth of penetration *d_avg_* is calculated by first measuring the area A_w_ and maximum width w_max_ of the wetted region for the two split sections using CAD software [35]. Then, *d_avg_* is taken as the average of the A_w_ divided by w_max_ for each half.

An increase in sample weight indicates that the samples were still hydrated or saturated throughout the test. The changes that occur in the samples throughout the test can be quantified by checking the measured inflow volumes and accurately measuring the sample weight before and after the test. Changes in the mass of samples during the test make it possible to determine the porosity, that is, the volume of pores that are filled with water, expressed as a percentage or fraction of the volume of concrete.

These samples were characterized by a much smaller mass and it was possible to apply an electronic balance with a resolution of 0.01 g, thereby significantly increasing the accuracy of the measurements. These results are shown in Table 3 and adapted to calculate the concrete permeability coefficient according to Equation (1).

### 2.3. Coefficient of Permeability Obtained by the Flow Pump Technique

The flow pump technique developed by Olsen et al. [21] was especially designed for low-permeability materials. Compared to the conventional constant- or falling-head methods, this technique allows permeability measurements to be obtained much more rapidly at the required gradients, even for very low permeable material.

A number of constant head permeability tests were carried out according to ASTM D2434 [43]. Tests were performed in a cell, at which the hydraulic gradient and confining pressure were applied and controlled. The pressure at the upper side of each sample was set as 0 MPa, while at the bottom, the pressure took values of 0.5 MPa. The confining pressure within the cell was equal to 0.6 MPa. The water permeability of each sample was calculated as the average of the last three measurements. The room temperature at which the tests were performed was equal to 22 °C. The permeability apparatus used during the test is shown in Figure 2.

The experimental setup shown in Figure 2 was designed for the water permeation tests of the concrete samples. For each sample, the testing procedure involved the measurement of the main experimental parameters, such as the dimensions of the specimen, the sample’s mass before and after the experiment, the driving pressure, the confining pressure, and the flow rate. Moreover, the experimental procedure for the permeability measurement ensured saturation of the sample material up to full saturation by the flow rate during the test.

The permeability cell is a type of triaxial cell originally developed to test the permeability of fine-grained soil or rock. The cell is able to withstand the confining pressures (up to 1.0 MPa) necessary for measuring flow in materials with permeability as low as 10^−15^ m/s. To avoid any lateral leakage and to ensure uniaxial flow, the circumference of the cylindrical specimen was covered with a rubber membrane that was sealed to the pedestal and upper cap with O-rings. The hydraulic head was applied using a volume–pressure controller. The driving pressure used in this study was 0.5 MPa. It has been found that the presence of compressive stress below a certain threshold value decreases the permeability, but when the applied stress exceeds this threshold, a significant increase in the permeability occurs [44,45,46]. Therefore, the confining pressure was 0.1 MPa higher than the driving pressure to avoid radial water flow. The pressure inside the concrete specimen was 0.5 MPa. The control and measurement of the pressure and flow rate were improved by using a syringe pump capable of providing stable and accurate flows as low as 0.00001 mL/min, which is an absolute requirement for measuring the permeability of low-permeability materials. The pump also provided a stable and precise pressure with an accuracy of 0.1% of the full scale (resolution of 1 kPa). 

The flow rate was determined from the volume of water flowing during an imposed specific time phase. The change in flow rate was recorded regularly, with the frequency of measurements depending on the type of material and the time elapsed since the beginning of the test. The readings were taken several times a day (collected during a given period). The volume of the water that passed through the sample in unit time for the test pressure was then calculated and expressed as the water permeability. When steady state conditions had been achieved at the test pressure, the inflow corresponding to this could be used to calculate the permeability coefficient. The initial and final masses of the samples were measured with a resolution of 0.01 g to determine their relative contents of pores filled with water.

Amriou and Bencheikh [47] noted that a shorter flow path results in a better steady state flow condition. Therefore, the permeability test can be carried out in a shorter time. Plotting the cumulative flow versus time curve indicated a nonlinear behavior in the beginning and then an almost linear one after 50 h, proving that the flow became steady. This is a quick method of determining the permeability of ordinary concrete. The first phase of the test (the nonlinear part of the curve) is related to continued saturation, the absorption of water into the interconnected pores, or continued hydration. The duration of this phase in other steady state water flow tests is 7–21 days, when the inlet flow becomes equal to the outlet flow in the sample [5,48,49]. In the present study, the water permeability was monitored for 3–8 days for all of the samples tested. 

The permeability coefficient k of the samples was determined using Darcy’s experimental law [7] (Equation (3)). In the calculations, a unidirectional and laminar flow of incompressible fluid, negligible inertial force, porous saturated material, and steady state equilibrium were assumed. The equation for calculating the permeability coefficient for cylindrical samples was obtained as:(3)$$k=\frac{QL}{F\Delta H}$$
where *k* is permeability coefficient (m/s), *Q* is the volumetric flow rate (m^3^/s), *L* is the length of the flow path (m), *F* is a cross-section of the sample exposed to water flow (m^2^), and Δ*H* is the drop in hydraulic potential difference (m).

## 3. Test Results

### 3.1. Values of the Concrete Permeability Coefficient Obtained by the Depth Penetration Method 

The average values of the measured water penetration depth and the calculated permeability coefficients of all of the different concretes and the coefficients of variation (CVs) are presented in Table 3. The CVs for all of the obtained values of permeability using Valenta’s formula ranged between 4.8 and 8.9%, and the overall average CV was approximately 7%. This confirms the very low variation of the test results. In the calculation of the permeability coefficient, as defined by the modified Valenta’s equation, three different variants were adopted: (1) by incorporating the volume of pores filled by the advancing water front; (2) by incorporating the ratio of average to maximum penetration depth; (3) by incorporating the apparent air content from the concrete mix. The obtained results showed slight differences in the measured values of the water penetration depth recorded for the particular types of tested concretes. In all cases, the average depth of water penetration was smaller than 50 mm and so the tested concretes (according to DIN 1045 [50]) should be considered watertight. Nevertheless, the beneficial effect of the polycarboxylate superplasticizer, which reduces the penetration range of water in concrete, is visible. In Figure 3, there are photos showing the water penetration depth for the tested specimens, which were split after testing.

### 3.2. Values of the Concrete Permeability Coefficient Obtained by the Flow Pump Technique at a Constant Level of Hydraulic Head

Water permeability is generally determined by accumulated flow and a permeability coefficient. The curves of the total flow or permeation were plotted on the basis of the change in the flow rate. Figure 4 demonstrates that the permeability test was conducted with a steady state of water flow in the samples. The curves clearly distinguish a nonlinear behavior in the first hours of the flow pump permeability test, and then a linear relationship, indicating a steady flow. The gradient of the permeability curve is proportional to the coefficient of permeability at any time [3]. Moreover, slight slope changes at low flow rates result in significant changes in the permeability (even in the order of magnitude).

The water permeability of saturated concrete is seldom the prevailing conveying mechanism in practical applications [9]. If non-saturated samples are tested, the time taken to establish the steady state will be longer than normal; hence, the duration of the test will be extended [7]. It is believed that the full saturation state is extremely difficult to obtain within a short time, especially for low-permeability materials [4]. Safiuddin and Hearn [51] found that vacuum saturation is the most efficient technique for measuring the permeable porosity of concrete and should therefore be recommended. El-Dieb and Hooton [10] noticed that the start time after which the permeability coefficient was measured increases with a decrease in permeability. For permeability coefficients of the order of 10^−13^ ÷ 10^−14^ m/s, the initializing times were approximately 140–150 h. Kameche et al. [8] reported that with the ordinary concrete used (highly permeable material), it took approximately 6 h to stabilize the flux through the test specimen.

The transient rise and transient decay phases of the presented tests in low-permeability concretes took a shorter period of time. This does not mean that when the stable permeation stage is reached in a shorter amount of time, the material is more permeated. Instead, it indicates that, before the tests began, the concretes also had a very high degree of cement hydration. The fluid–cement matrix interactions during measurements can significantly affect the permeability. The interactions of the water–cement matrix could be reduced in this method because of the continuing hydration of mature cement paste in a relatively short time required to carry out the measurement. Nokken and Hootoon [6] noticed that a somewhat lower permeability could be obtained since the samples were undoubtedly further hydrated during the measurement. Moreover, hydration can change the pore structure, which could lead to changes in permeability only during a long testing time, especially in the early hydration stage [4].

Different effects of the applied admixtures were also observed, as shown in Figure 4. Admixture (A), which was used to improve the compressive strength, caused a higher cumulative flow, while the cumulative flow of the superplasticizer (B) was smaller. The interaction of the two admixtures had the advantage of reducing the cumulative flow and, thus, the permeability of the concrete. The reason for this is that the concrete with admixture (A) had higher porosity, which allowed a larger cumulative flow and permeation. The introduction of an additional admixture (B) caused a significant reduction in the porosity and, consequently, a decrease in the permeability. 

In order to compare the results obtained from this method with those obtained from the depth penetration method, the coefficient of permeability was evaluated as an average value of 71 and 73 h. In Figure 5, a typical permeability coefficient curve in time is presented. Each point in the figure is a value of the permeability coefficient calculated according to Equation (3) for two successive readings.

The coefficient of permeability decreased with time. The reduction in the permeability coefficient value over time was not due to the continuous hydration of the cement. During a long testing period, even small temperature variations cause appreciable fluctuations in the response of the rate flow versus time because the high accuracy measurements of flow water volume. These fluctuations are caused by the thermal expansion and contraction of both the equipment and water for the permeability test [15]. In Figure 5, the scattering of the points around 72 h after the beginning of the test are marked on the dotted line. The final result was calculated as an average of three points and shows a non-significant difference in values. During the test, the permeability coefficients of the samples decreased for 50 h from the beginning of the test and then decreased gradually as the permeation continued. Next, the decrease in the permeability coefficients decelerated and reached a stable value.

A summary of the results for each concrete is given in Table 4. The results presented are from single experiments. The values of the permeability coefficient were calculated by Equation (3). The relative content of the open pores that were filled with pressurized water was determined based on the change in the mass of the samples during the permeability tests.

## 4. Discussion

In the proposed direct method of determining concrete permeability, the average permeability coefficient values in a given unit of time (~72 h) were assumed. The results of the tests were affected by solar gains in the room and a diurnal temperature variation on the outside. Therefore, visible cyclic scattering of points in time is visible in Figure 5. To ensure pressure stability in the permeability measuring device, the room temperature must be controlled. This procedure is of great importance because it eliminates difficulties in interpreting the results and allows their unambiguous evaluation, as noted in the work of Zhang et al. [16]. However, slight temperature fluctuations in the room do not affect the cumulative amount of water flow versus the cumulative time (Figure 4). The concrete permeability test results obtained using a syringe pump at a constant level of hydraulic head were taken as reference values. The values found are comparable to those of other studies, despite some variations [7,35]. A direct relationship between porosity and permeability has been introduced by some researchers [4,38,52,53,54,55,56,57,58]. The relationship between the permeability coefficient value and the porosity of the concretes is illustrated in Figure 6. The obtained characteristics of concrete permeability at a constant W/C ratio are in agreement with the literature data for cement materials, despite the application of different test methods to determine both the permeability and the porosity (e.g., mercury intrusion porosimetry (MIP)). The tested materials showed a narrow range of values of both porosity and permeability in comparison to the cited ones [59]. Recording and showing such small changes in both parameters proves the accuracy of the applied testing techniques (e.g., automatic pycnometers and flow pump technique).

It can be seen from Figure 6 and Table 2, in combination with Table 4, that the experimental results fit very well, with the correlation coefficient (r) being greater than 0.99. The results clearly show that the permeability coefficient values of the studied concretes increased as the porosity increased. This behavior could be explained by two factors: (i) the action of admixture (A) caused a significant increase in porosity, but the pore size remained small due to the clogging of the capillary pores, and despite this, the permeability increased; (ii) the action of admixture (B) caused a small increase in porosity, but together with admixture (A), the increases in porosity were negligible, resulting in a highly impermeable concrete. These results are reliable and explain the considerable influence of the admixtures on the water permeability of concretes.

The results from the flow pump indicate significantly lower permeability values, up to an order of magnitude lower than the calculated values based on the penetration depth (Figure 7).

The coefficients of permeability for all of the concretes changed in a similar way, although they concerned materials of different porosities and compressive strengths. Generally, the values were different by approximately an order of magnitude, but even more than two orders of magnitude if correction was applied taking into account the average to maximum penetration depth ratio. The values obtained on the basis of the modified Valenta Equation (1) by incorporating the apparent air content are interesting, as this approach can easily be applied in practice. A better agreement with the reference data in this case was found using the following formula:(4)$$k=\left(\frac{\varepsilon }{10}\right)\frac{d_{\max}^{2}}{2Ht}$$
where *ε* is the apparent air content in concrete mix.

The favorable agreement affords confidence in the validity of the developed method. The average fitting error of the data calculated from Equation (4) compared to the reference values was only 59%.

## 5. Conclusions

Two different methods for evaluating the permeation property of tested concretes were applied: The flow pump technique and the depth of penetration method. The experimental results confirmed that the depth penetration test and the flow pump test at a constant hydraulic head result in similar values of the coefficient of permeability when similar samples are tested under the same conditions.

For the first time, a novel direct method of measuring the in-flow of cement-based materials was applied based on the application of a constant head and measuring an extremely low flow rate. The flow pump technique is better suited to automatic data acquisition and is more reliable and sensitive than conventional methods. The results for the cumulative water flow versus time were plotted to determine flow at steady state. Next, the steady flow rate was determined and the coefficient of the water permeability of concrete was calculated using Darcy’s law.

The fundamental advantages are that the flow pump method is very fast, highly accurate, and simple enough to apply. The greatest dynamics of the changes in the flow of water under constant pressure were observed during the first day. The second day was a transitional stage. Next, there was a very small decrease in the permeability coefficient corresponding to the quasi steady state conditions. The tests performed confirmed that the three days of measurements guarantee the determination of the permeability coefficient in weakly permeable concrete. It can be expected that in the case of more permeable concretes, a shorter time would be needed to reach steady state conditions. The results obtained by the water penetration depth method were adequate for such materials. Moreover, a novel indirect method of determining concrete permeability based on the modified Valenta equation used in the depth of water penetration tests was proposed. These results confirmed that the proposed methods can be used for evaluating the water permeability of concrete. Based on the experimental data provided, the presented methods appear to have merit in determining water permeability and are capable of properly and reliably identifying the influence of the properties of concrete on permeability.

This paper also presented experimental results concerning the influence of a new generation of polycarboxylate superplasticizers and a chemically active admixture to increase the compressive strength and to improve the other properties of concrete, but it was not a main goal of this research.

To summarize, it should be emphasized that the proposed flow pump method of testing the water permeability of concrete has not been previously used in scientific research and has the advantage that it can be successfully used in practice. The tests were carried out on concrete samples prepared in the laboratory, but the flow pump technique is suitable for testing concrete from structures. It is common knowledge that concrete made according to the same recipe in situ (in engineering constructions) shows different quality characteristics from that of concrete cured in a laboratory. A lot of work on this subject has been carried out already, in which field research plays an important role. In order to determine the permeability of concrete in a structure, it is sufficient to take core samples of a small height (approximately 100–150 mm) and a small diameter of 100 mm or more at designated points. Shallow boreholes with a small diameter will disturb the structure to a minimal extent, and concrete losses can be easily filled in. Drilling can be done in sections without reinforcement. However, it is also possible to test samples containing horizontal reinforcement (vertical along the height of the sample could be a privileged flow path). The proposed precise and fast way of measuring water permeability (at least 24 h) provides a new perspective on concrete testing in structures compared to the known indirect methods such as water absorption, which are very time-consuming.

## Figures and Tables

**Figure 1 materials-13-05624-f001:**
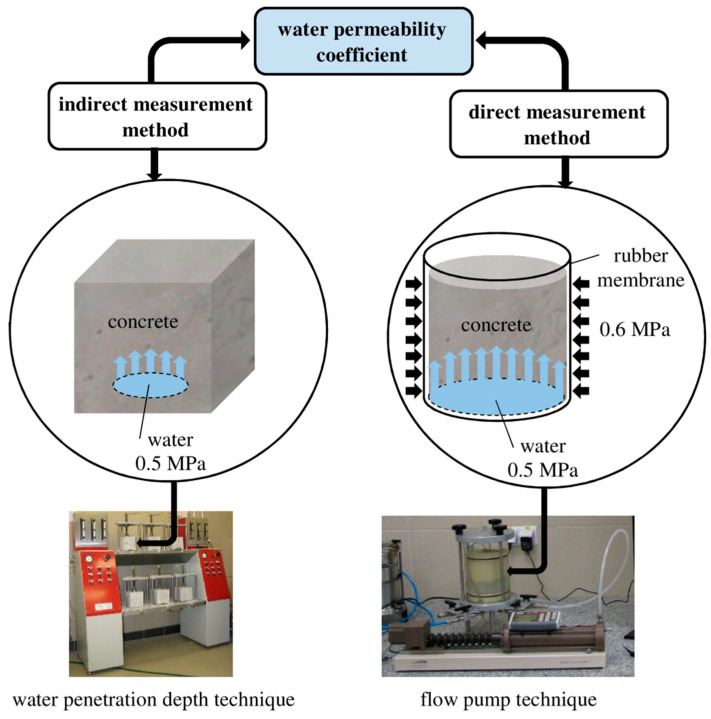
Scheme of the indirect and direct methods of testing the permeability coefficient.

**Figure 2 materials-13-05624-f002:**
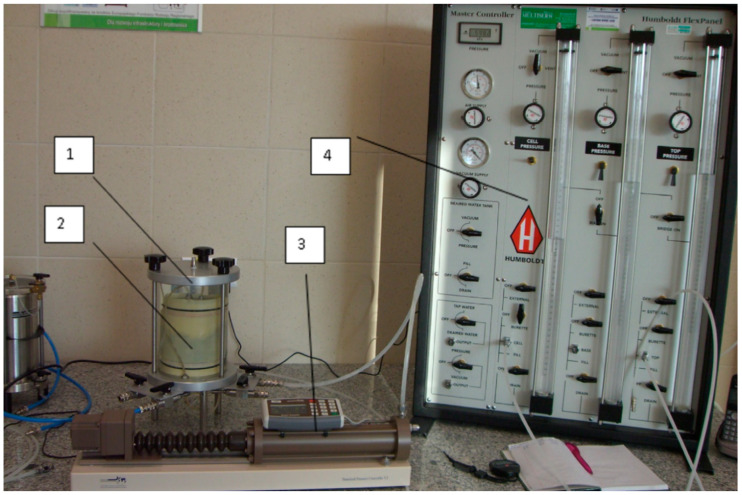
Photos of the permeability apparatus: (**1**) Testing cell; (**2**) specimen; (**3**) volume/pressure controller; (**4**) cell pressure controller.

**Figure 3 materials-13-05624-f003:**
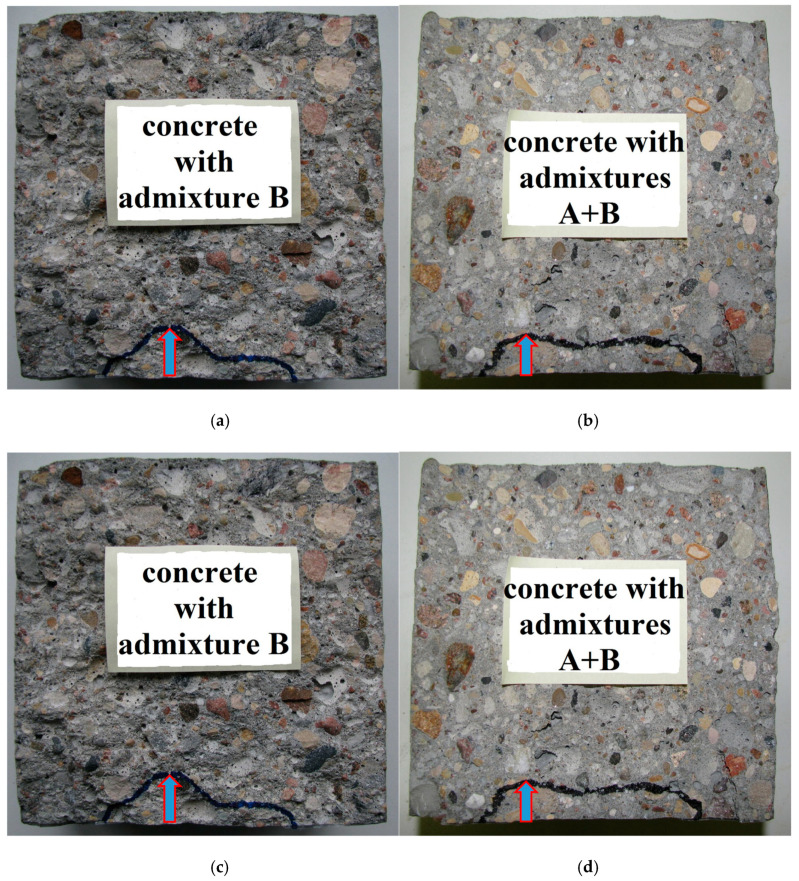
Cross-sections of the chosen tested specimens, which were split after testing, showing the water penetration depth of the tested specimens for the different types of concrete. (**a**) concrete without admixture, (**b**) concrete with admixture A, (**c**) concrete with admixture B, (**d**) concrete with admixture A + B.

**Figure 4 materials-13-05624-f004:**
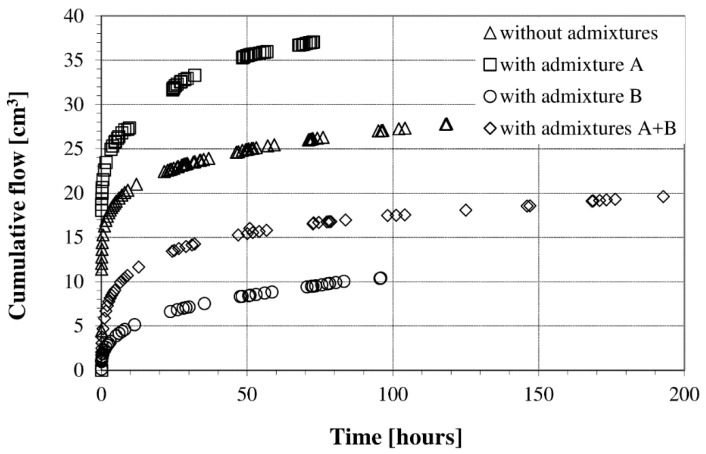
Dependence of the cumulative flow with the used admixtures at a pressure of 0.5 MPa.

**Figure 5 materials-13-05624-f005:**
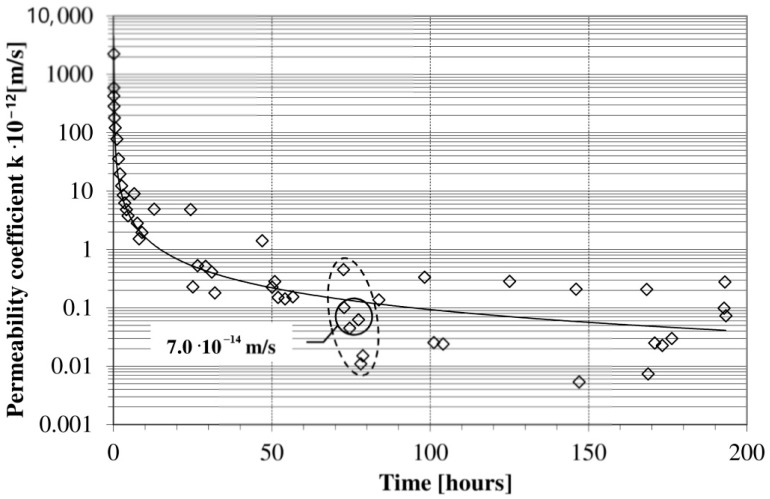
Typical permeability coefficient curve in time. Concrete with the A and B admixtures under a hydraulic pressure of 0.5 MPa.

**Figure 6 materials-13-05624-f006:**
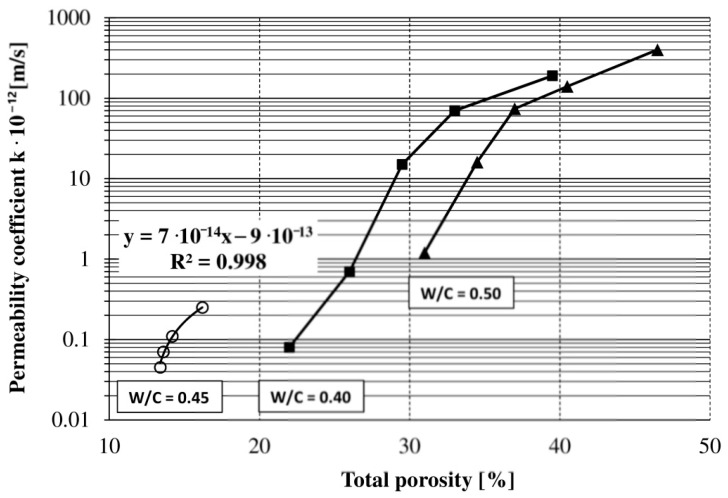
Comparison of the relationship between permeability and porosity for three W/C ratio samples. The measured porosities in the experiments for W/C 0.45 were obtained with two automatic pycnometers. The collection of data regarding permeability as a function of porosity for W/C 0.4 and 0.5 was obtained with mercury intrusion porosimetry (MIP).

**Figure 7 materials-13-05624-f007:**
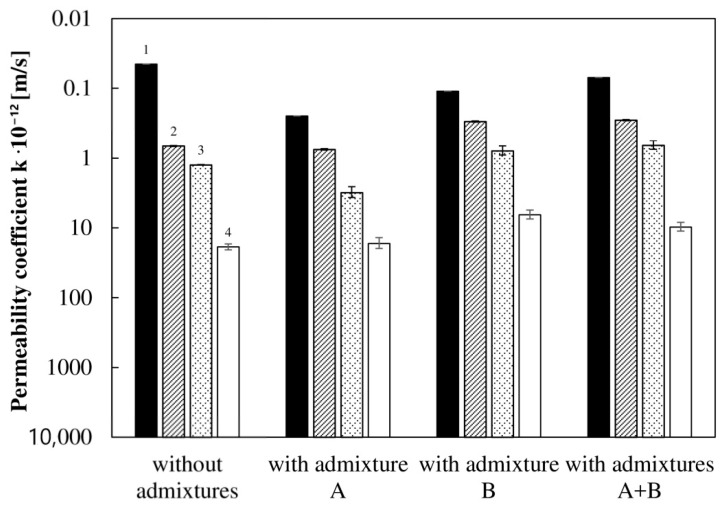
Comparison of the permeability of the concretes with different admixtures obtained from two methods: (1) the flow pump test method at a constant hydraulic head and (2–4) the depth of water penetration test method—where 2 is corrected by the content of pores filled with water, 3 is corrected by the apparent air content, and 4 is corrected by the relationship between the average and maximal penetration depths. Error bars indicate the standard deviation of the experiments (2σ).

**Table 1 materials-13-05624-t001:** The composition of the concrete mixture at a scale of 1 m^3^.

Component	Content
Cement CEM I 32.5 R (C), kg	300
Water (W), dm^3^	135
Sand 0–2 mm, kg	713
Gravel 2–16 mm, kg	1323
W/C A/C (% by cement weight)	0.45 1.6
B/C (% by cement weight)	0.8

Note: W/C, water-cement ratio; A/C, percentage of admixture A by cement weight; B/C, percentage of superplasticizer B by cement weight.

**Table 2 materials-13-05624-t002:** The obtained test results for the concrete mixes and the hardened concretes.

Type of Material	Concrete Mix		Hardened Concrete			
	Density (kg/m^3^)	Apparent Air Content (%)	Total Porosity (%)	Coefficient of Variation *n* * = 6 (%)	Concrete Compressive Strength at 28 Days (MPa)	Coefficient of Variation *n* = 4 (%)
Without admixture	2367	2.8	13.4	0.3	39.8	3.0
With admixture A	2306	5.4	16.2	0.3	47.8	2.2
With superplasticizer B	2341	4.2	14.2	0.4	44.0	3.5
With admixtures A + B	2343	3.2	13.6	0.3	47.4	2.8

* *n*, number of tests.

**Table 3 materials-13-05624-t003:** The test results of the water penetration depth and the calculated values of the permeability coefficient.

Type of Concrete	MWPD (cm)	CV *n* = 3 (%)	AWPD (cm)	Permeability Coefficient			CV *n* = 3 (%)
				vpawf (m/s)	a/mpd (m/s)	aac (m/s)	
Without admixture	3.4	2.4	2.2	6.69 × 10^−13^	1.88 × 10^−11^	1.25 × 10^−12^	4.8
With admixture A	3.9	4.4	2.1	7.51 × 10^−13^	1.68 × 10^−11^	3.12 × 10^−12^	8.9
With admixture B	2.2	3.7	1.3	2.99 × 10^−13^	6.51 × 10^−12^	7.85 × 10^−13^	7.4
With admixtures A + B	2.3	3.5	1.6	2.86 × 10^−13^	9.73 × 10^−12^	6.54 × 10^−13^	7.1

Note: MWPD, maximum water penetration depth; CV, coefficient of variation; AWPD, average water penetration depth. Permeability coefficient incorporated by: vpawf, volume of pores filled by the advancing water front; a/mpd, the ratio of average to maximum penetration depth; aac, the apparent air content from the concrete mix.

**Table 4 materials-13-05624-t004:** Results of the permeability test (from the modified flow pump method at a constant level of hydraulic head).

Type of Concrete	Water Pressure (MPa)	Pressure Duration Time (hours)	Relative Content of Pores Filled with Water (%)	Total Porosity (%)	Permeability Coefficient (m/s)
Without admixture	0.5	72	1.5	13.4	4.50 × 10^−14^
With admixture A	0.5	72	1.3	16.2	2.5 × 10^−13^
With superplasticizer B	0.5	72	1.6	14.2	1.1 × 10^−13^
With admixtures A + B	0.5	72	1.4	13.6	7.0 × 10^−14^

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
