# Peer review of "An Experimental Assessment of the Water Permeability of Concrete with a Superplasticizer and Admixtures"

_materials, 2020, doi:10.3390/ma13245624_

Round 1
Reviewer 1 Report
This paper reported experimental investigation on water permeability of concrete.
1) Recently, non-destructive testing method measuring permeability (water absorption) has been proposed to evaluate properties of real concrete structures as followings. The authors are recommended to introduce them and/or other studies in the introduction part to emphasize importance and future possibility of permeability measurements.
1a) M.H. Nguyen, K. Nakarai, Y. Kubori, S. Nishio, Validation of simple nondestructive method for evaluation of cover concrete quality, Constr. Build. Mater. 201 (2019) 430–438.
2) Literature view on pumping should be added in the introduction part. For example, the following paper may help the authors.
2a) Huajian Li, Deyi Sun, Zhen Wang, Fali Huang, Zhonglai Yi, Zhengxian Yang, Yong Zhang, A Review on the Pumping Behavior of Modern Concrete, Journal of Advanced Concrete Technology, 2020
3) Literature view on polycarboxylate superplasticizer should be added in the introduction part.
4) Curing conditions shown in lines 106-108 are not clear. What is “preparation” in the explanation? Is it casting of specimens?
5) Degree of saturation may affect the results of water penetration depth with EN 12390-8 because the tests do not request pre-saturation. In 2.2, explain the specimen conditions for the water penetration tests.
6) CV in Table 3 is very small (2.4-4.4%). In general, water penetration depth shows larger variation because of special heterogeneity with aggregate. Show example of photos showing water penetration depth with small variation.
7) Huge studies have reported good relation between total porosity and permeability coefficient same as the results in Fig.5 (not Fig. 4. Correct line 341). The reviewer recommends the authors to add results from the literature to verify their results in this study.
Reviewer 2 Report
The novelty of the paper is not really well-addressed. A comparison between different methods for measuring permeability is considered, but not innovation incorporates in the methods and not significant discussion is included. Authors consider the assessment of two different additives, but many uncertainties related to the use of the additives are observed.
Polycarboxylates are considered water-reducing additives; however, authors maintain the w/c ration and the cement content constant. The mortar obtained will be affected by the presence of the polycarboxylate in the fresh-state, but no characterization of the fresh state is considered in the manuscript.
The additive (A) is supposed to increase the compressive strength and sealing by filling of capillary pores with low soluble polycarboxylate. However, an increase of porosity is observed in the mortar incorporating this additive, probably due to the design of the mortar (also this type of additives can influence on the properties of the fresh state, and different w/c ratio would be consider).
In fact, authors do not give any explanation to the fact that the different mortars with additives promotes and increase of porosity and also an increase on the concrete compressive strength at 28 days...
In figure 5 it is not clear which data are represented as authors mentions that lesser total porosity is related to lesser permeability coefficient. However, the mortar with lesser porosity is the reference one but the reference mortar is not the one with lesser permeability coefficient.
In my opinion the study should be revised including results and analysis of the fresh state of the different mixes and the quality of the manuscript must be increased before publishing in Materials Journal.
Reviewer 3 Report
This paper is a very interesting article dealing with the water permeability test method for concrete.
First of all, I think that this title does not reflect the content of this paper because the effect of admixtures on the permeability of concrete was evaluated only marginally with respect to the focus on the test methods.
the introduction is complete even if some references reported at the end of the paper is not cited within the text (see for example the interesting ref 23).
in the section "materials and methods" some results are reported. this is not a good idea from the point of view of the reader. in fact, the text in some part is difficult to read. maybe a revision of the partition in paragraphs is needed, especially in chapter related to materials&methods and results.
on the contrary, the discussion seems to be reasonable.
Concerning the permeability tests according to EN 12390-8 on concretes manufactured with waterproofing admixtures (i.e. admixture A), I want to highlight that results reported in other papers (i.e.
Pazderka et al. “The Speed of the Crystalline Admixture’s Waterproofing Effect in Concrete.” Key Engineering Materials, vol. 722, 2016, pp. 108–112.
Coppola et al. "Innovative carboxylic acid waterproofing admixture for self-sealing watertight concretes", 2018, ConBuildMat 171, 817-824))
show an interesting improvement in watertightness of concrete regardless of Superplasticizer content.
